# Genetic differentiation of a southern Africa tepary bean (*Phaseolus acutifolius* A Gray) germplasm collection using high-density DArTseq SNP markers

Saul Eric Mwale[1,2]*, Hussein Shimelis[1], Wilfred Abincha[3], Wilson Nkhata[1,4], Abel Sefasi[5], Jacob Mashilo[1]

1 School of Agricultural, Earth and Environmental Sciences, African Centre for Crop Improvement (ACCI), University of KwaZulu-Natal, Pietermaritzburg, South Africa, 2 Biological Sciences Department, The African Centre of Excellence in Neglected and Underutilized Biodiversity (ACENUB), Mzuzu University, Luwinga, Mzuzu, Malawi, 3 Kenya Agricultural and Livestock Research Organization (KALRO), Non-Ruminant Research Institute, Kakamega, Kenya, 4 Alliance of Bioversity International Institute of Tropical Agriculture (CIAT), Chitedze Agricultural Station, Lilongwe, Malawi, 5 Lilongwe University of Agriculture and Natural Resources, Lilongwe, Malawi

* mwale.s@mzuni.ac.mw

**Data Availability Statement:** All relevant data are within the paper and its Supporting information files.

## Abstract

Genetic resources of tepary bean (*Phaseolus acutifolius* A. Gray) germplasm collections are not well characterized due to a lack of dedicated genomic resources. There is a need to assemble genomic resources specific to tepary bean for germplasm characterization, heterotic grouping, and breeding. Therefore, the objectives of this study were to deduce the genetic groups in tepary bean germplasm collection using high-density Diversity Array Technology (DArT) based single nucleotide polymorphism (SNP) markers and select contrasting genotypes for breeding. Seventy-eight tepary bean accessions were genotyped using 10527 SNPs markers, and genetic parameters were estimated. Population structure was delineated using principal component and admixture analyses. A mean polymorphic information content (PIC) of 0.27 was recorded, indicating a relatively low genetic resolution of the developed SNPs markers. Low genetic variation (with a genetic distance [GD] = 0.32) existed in the assessed tepary bean germplasm collection. Population structure analysis identified five sub-populations through sparse non-negative matrix factorization (snmf) with high admixtures. Analysis of molecular variance indicated high genetic differentiation within populations (61.88%) and low between populations (38.12%), indicating high gene exchange. The five sub-populations exhibited variable fixation index ($F_{ST}$). The following genetically distant accessions were selected: Cluster 1:Tars-Tep 112, Tars-Tep 10, Tars-Tep 23, Tars-Tep-86, Tars-Tep-83, and Tars-Tep 85; Cluster 3: G40022, Tars-Tep-93, and Tars-Tep-100; Cluster 5: Zimbabwe landrace, G40017, G40143, and G40150. The distantly related and contrasting accessions are useful to initiate crosses to enhance genetic variation and for the selection of economic traits in tepary bean.

**Funding:** Yes. Kirkhouse Trust SCIO provided the funding for the genotyping of tepary bean germplasm.

**Competing interests:** No The authors have declared that no competing interests exist.

## Introduction

Tepary bean (*Phaseolus acutifolius* A. Gray) is a climate-smart legume crop that provides food and nutrition security in arid and semi-arid regions of the world [1]. It is a self-pollinating diploid (2n = 2x = 22) species with a genome size of approximately 647 million base pairs (Mbp) [2]. The grains are a source of proteins, lipids and essential mineral elements [3]. The crop is tolerant to drought and heat stress [1,4,5], allowing its cultivation in dry environments. It is highly resistant to diseases including bacterial blight [*Xanthomonas campestris pv. Phaseoli*] [6], *Fusarium* wilt (*Fusarium oxysporum*) [7], and bean golden mosaic virus [8]. As a result, the crop served as a useful gene donor for disease-resistance breeding in common bean [9,10].

In Southern Africa, tepary bean is mostly cultivated by smallholder farmers in Botswana, South Africa, Zimbabwe, Zambia, and Malawi using genetically unimproved landrace varieties [11,12]. This is attributed to limited breeding efforts to develop improved varieties with farmer, consumer and market-preferred traits [13]. The crop yield in the region is approximately 500 kilograms per hectare [14,15] compared to the potential yield of greater than 2000 kilograms per hectare [16]. To address the imminent challenge of low productivity in Southern Africa, the University of KwaZulu-Natal's African Center for Crop Improvement (ACCI) acquired a diverse germplasm panel of tepary bean from the International Centre for Tropical Agriculture (CIAT-Columbia) [17]. The collected genetic resources of tepary bean are yet to be explored using dedicated genomic resources. Reportedly, genetic analysis using various molecular markers developed for common bean (*Phaseolus vulgaris* L.) such as simple sequence repeat (SSR), random amplified polymorphic DNA (RAPD), amplified fragment length polymorphism (AFLP), and sequenced characterized amplified region (SCAR) revealed a low to moderate genetic variation when applied in tepary bean [18–25]. There is a need to assemble genomic resources specific to tepary bean for germplasm characterization, selection, and breeding.

The reference genome to which the molecular marker covers the genome influences germplasm characterization accuracy and predictability [26]. Single nucleotide polymorphism (SNP) markers are widely abundant across the genome and quickly unravel diversity; hence, they are the most preferred for diversity and association mapping studies [27,28]. Gene-based SNP markers, which mainly uncover functional variation, were used to assess the population structure and genetic diversity of wild and cultivated tepary bean genotypes using the common bean reference genome [26]. However, the genotyping assay was limited in capturing several loci (< 60%) in tepary bean since it was aligned to the common bean reference genome [26]. A draft tepary bean reference genome was developed recently, providing enormous opportunities for developing tepary bean genomic resources [1]. Genotyping-by-sequencing (GBS) is one of the most cost-effective approaches for concurrent SNPs identification and genotyping [29]. Diversity Array Technology Sequencing (DArTSeq) is a GBS platform that combines the principles of genome complexity reduction methods with high throughput sequencing. It allows the simultaneous identification of numerous SNPs across a genome at an affordable cost [30]. DArTseq SNP markers have been used for quantitative trait loci (QTL) mapping, genome-wide association studies, the development of linkage maps, genetic diversity, and population structure analyses in several legume crops, including pigeonpea, chickpea, soybean, and common bean [30–34]. Genetic diversity and population structures were examined in Interspecific Mesoamerican X Wild Tepary (IMAWT) population using GBS with high-density SNP markers aligned to the common bean reference genome [5]. Further, GBS with high-density SNP markers mapped to the tepary bean reference genome in a tepary bean diversity panel has been used to determine the extent of genetic diversity and population structure [35]. Nevertheless, the southern Africa tepary bean diversity panel collection, comprising landraces,

released lines, and breeding lines, has not yet been characterized with high-density DArTSeq SNP markers matched to the tepary bean reference genome. Genetic analysis of the southern Africa tepary bean germplasm collection could potentially capture a broader range of genetic variation and provide useful information for heterotic grouping, genome-wide association mapping, marker-assisted selection for precision, and speed breeding. Therefore, the objectives of this study were to deduce the genetic groups in tepary bean germplasm collection using high-density DArT-based single nucleotide polymorphism markers and select contrasting genotypes for breeding.

## Materials and methods

### Plant genetic materials

A panel of 78 tepary bean germplasm collection comprising released varieties, breeding lines, and landraces were used for the study. The accessions were sourced from the International Center for Tropical Agriculture (CIAT)/Colombia, the United States Department of Agriculture (USDA), and farmers in South Africa and Zimbabwe. The names of the genotypes and their sources of origin are presented in "Table 1".

### DNA extraction and SNP genotyping

The above tepary bean germplasm were genotyped at SEQART Africa in the International Livestock Research Institute (1.2693˚ S, 36.7216˚ E) in Nairobi. Genomic DNA was extracted using the TANBEAD Plant Extraction Kit (Diagnocine, Hackensack, NJ, USA). The genomic DNA extracted was in the range of 50–100 ng/ul. DNA quality and quantity were checked on 0.8% Agarose Gel. The genomic DNA was subjected to restriction digestion using Mst1 and Pst1 as rare and frequent cutters, respectively. Ligation of the digested DNA fragments was accomplished by both common and barcode adapters. This was followed by selective amplification via polymerase chain reaction (PCR) of the adapter-ligated fragments. Pooling and purification of the PCR products were accomplished through a QIAquick PCR purification kit (QIAGEN GmbH, Hilden, Germany). An Illumina Hiseq 2500 (Macrogen, Seoul, Korea) that utilizes single reads was used to sequence the purified PCR products. DNA libraries were constructed according to Kilian et al. [36]. DArTseq marker scoring was achieved using DArTsoft 14, an in-house marker scoring pipeline based on algorithms [37–39]. Two types of DArTseq markers were scored: silica-DArT markers and SNP markers, which were both scored as binary for the presence (1) or absence (0) of the restriction fragment with the marker sequence in the genomic representation of the sample. 11, 318 SNP markers were aligned to 11 chromosomes of the reference genome of *P. acutifolius*_580_v1.0 to identify chromosome positions [1]. The markers used in the current study were highly reproducible, with polymorphic information content (PIC) that varied from 0.01 to 0.05 and a mean call rate of 0.93 ranging from 0.81 to 1.00.

   **SNP quality control.** Quality control was implemented for the SNP data using the *raw.data* function of the *snpReady* package [40] in R statistical software version 4.3.1 [41]. SNP markers were filtered by eliminating markers with no chromosomal position and a minor allele frequency (MAF) of <5%. The percent similarity among identical markers was assessed, and markers with the highest missing data were eliminated. A total of 10527 markers with a SNP call rate greater than 95% were retained and used for genetic and population structure analysis. The *MVP.report* function of rMVP package in R was used to develop SNP density plot of the filtered SNPs [42]. Previously this analysis was done using CMplot package which has now been integrated in rMVP package.

**Table 1. Names and origins of tepary bean germplasm collection used in the study.**

| Genotype Code | Genotype name or designation | Origin | Genotype Code | Genotype name or designation | Origin | Genotype Code | Genotype name or designation | Origin |
|---|---|---|---|---|---|---|---|---|
| G1 | G40001 | Mexico | G27 | G40129 | Mexico | G55 | TARS-TEP-32 | USDA |
| G2 | G40005 | El Salvador | G28 | G40132 | Mexico | G56 | TARS-TEP-49A | USDA |
| G3 | G40013 | Nicaragua | G29 | G40133 | Mexico | G57 | TARS-TEP-49B | USDA |
| G4 | G40014 | Nicaragua | G30 | G40134 | Mexico | G58 | TARS-TEP-51 | USDA |
| G5 | G40017 | El Salvador | G31 | G40135 | Mexico | G59 | TARS-TEP-52 | USDA |
| G6 | G40019 | Mexico | G32 | G40136 | Mexico | G60 | TARS-TEP-54 | USDA |
| G7 | G40020 | Mexico | G33 | G40137 | Mexico | G61 | TARS-TEP-58A | USDA |
| G8 | G40022 | USA | G34 | G40138 | Mexico | G62 | TARS-TEP-58B | USDA |
| G9 | G40023 | USA | G35 | G40139 | Mexico | G63 | TARS-TEP-60 | USDA |
| G10 | G40031 | Mexico | G36 | G40140 | Mexico | G64 | TARS-TEP-64 | USDA |
| G11 | G40032 | Guatemala | G37 | G40143 | Mexico | G65 | TARS-TEP-73 | USDA |
| G12 | G40033 | Mexico | G38 | G40144A | Mexico | G66 | TARS-TEP-77 | USDA |
| G13 | G40035 | Mexico | G39 | G40145 | Mexico | G67 | TARS-TEP-83 | USDA |
| G14 | G40036 | Mexico | G40 | G40147 | Mexico | G68 | TARS-TEP-85 | USDA |
| G15 | G40042 | USA | G41 | G40148 | Mexico | G69 | TARS-TEP-86 | USDA |
| G16 | G40059 | El Salvador | G42 | G40150 | Mexico | G71 | TARS-TEP 23 | USDA |
| G17 | G40062 | Nicaragua | G43 | G40157 | Mexico | G72 | PI-310801 | Nicaragua |
| G18 | G40063 | USA | G44 | G40158 | Mexico | G73 | G40119 | Mexico |
| G19 | G40065 | USA | G45 | Zimbabwe landrace | Zimbabwe | G74 | PI-440786 | USDA |
| G20 | G40066A | USA | G46 | G40173A | Mexico | G75 | G40200 | Costa Rica |
| G21 | G40068 | USA | G48 | G40201 | Costa Rica | G79 | TARS-TEP97 | USDA |
| G22 | G40069 | USA | G49 | G40237 | Mexico | G80 | TARS-TEP112 | USDA |
| G23 | G40084 | Mexico | G50 | Uchokwane | South Africa | G81 | TARS-TEP101 | USDA |
| G24 | G40111 | Mexico | G51 | SONORA | Sonora | G83 | TARS-TEP93 | USDA |
| G25 | G40125 | Mexico | G53 | TARS-TEP-10 | USDA | G84 | TARS-TEP51 | USDA |
| G26 | G40127 | Mexico | G54 | TARS-TEP-22 | USDA | G85 | TARS-TEP100 | USDA |

USDA = United States Department of Agriculture.

## Genetic diversity and population structure analysis

Genetic diversity analyses were conducted to determine the polymorphic information content (PIC), minor allele frequency (MAF), observed heterozygosity ($H_o$), genetic diversityGD), fixation index, additive variance and dominance variance. The genetic parameters were computed using the *snpReady* package [40] utilizing the popgen function in R statistical software [41]. A Landscape and Ecological Associations (LEA) R package was used to determine the population genetic structure [43–45]. Using *snmf* function of LEA, ancestry coefficients were estimated where Cross-entropy criteria was used to identify the optimum number of K populations where K = 1 to 10 with six repetitions [46,47]. Principal component analysis was implemented in LEA package using *pca* function to validate the identified subpopulation through the elbow method [48]. The individuals in the subpopulations were identified using the developed Q matrix in R statistical software [41]. Admixture plot of individuals were plotted using the

developed Q matrix using *plot* function of base R. The results obtained from population analysis were subjected to Principal Coordinate Analysis (PCoA) and analysis of molecular variance (AMOVA) using the poppr package [49] in R statistical software [41]. Neighbor joining hierarchical cluster analysis was performed in R statistical software using a dendextend package [41].

## Results

### Marker characterization

Genetic parameters derived by SNP markers are presented in "Table 2". PIC ranged from 0.10 to 0.40, with a mean of 0.27. Most SNP markers were found within the following PIC categories; 0.2 to 0.03 followed by 0.3–0.4 and the least was 0–0.1 ("Fig 1"). The mean Ho and He were 0.30 and 0.32, respectively. The mean Va and Vd were 3415.19 and 1227.16, respectively. Markers revealed a mean effective population size (Ne) of 488.28. The majority of the SNP markers had minor allele frequencies ranging from 0.1 to 0.2 and the least MAF ranging from 0.4 to 0.5. The SNP markers that explained the most genetic diversity were within the category 0.2 to 0.4 and the least informative were between 0.0 to 0.1 ("Fig 1"). The SNP markers were widely distributed across the 11 chromosomes with a SNP density ranging from 0 to > 65 ("Fig 2").

### Population structure of the assessed tepary bean germplasm

Population structure analyses based on cross-entropy criteria revealed that the population was optimally structured into 5 sub-populations (K = 5) ("Fig 3"). The result was further validated by observing the principal component screen plot, which showed five principal components (PCs). Results from admixture analysis revealed the presence of genetic admixture across the five subpopulations ("Fig 4").

The mean coefficient of inbreeding (F) for the five subpopulations was 0.08 and variable among the populations ("Fig 5"). Further, the extent of genetic divergence was highest between subpopulations 1 and 5 (0.23), subpopulations 1 and 2 (0.21), subpopulations 3 and 5, and moderate for subpopulations 1 and 3 (0.08) ("Table 3").

### Analysis of molecular variance

Analysis of molecular variance revealed a 38.12% variation between the population and 61.87% within population ("Table 4"). This was further visualized using principal coordinate

**Table 2. Genetic parameters computed from genetic diversity assessment of tepary bean germplasm using high-density SNP markers.**

| Genetic parameter | Overall Mean | Minimum | Maximum |
|---|---|---|---|
| GD | 0.32 | 0.10 | 0.50 |
| PIC | 0.27 | 0.09 | 0.38 |
| MAF | 0.22 | 0.05 | 0.5 |
| Ho | 0.30 | 0.06 | 0.81 |
| He | 0.32 | 0.10 | 0.50 |
| Va | 3415.19 | | |
| Vd | 1227.16 | | |
| Ne | 488.28 | | |

GD = Genetic Diversity; PIC = Polymorphic information content; MAF = Minor allele frequency; Ho = observed heterozygosity; He = Expected heterozygosity;

Va = additive variance; Vd = dominance variance; Ne = effective population size.

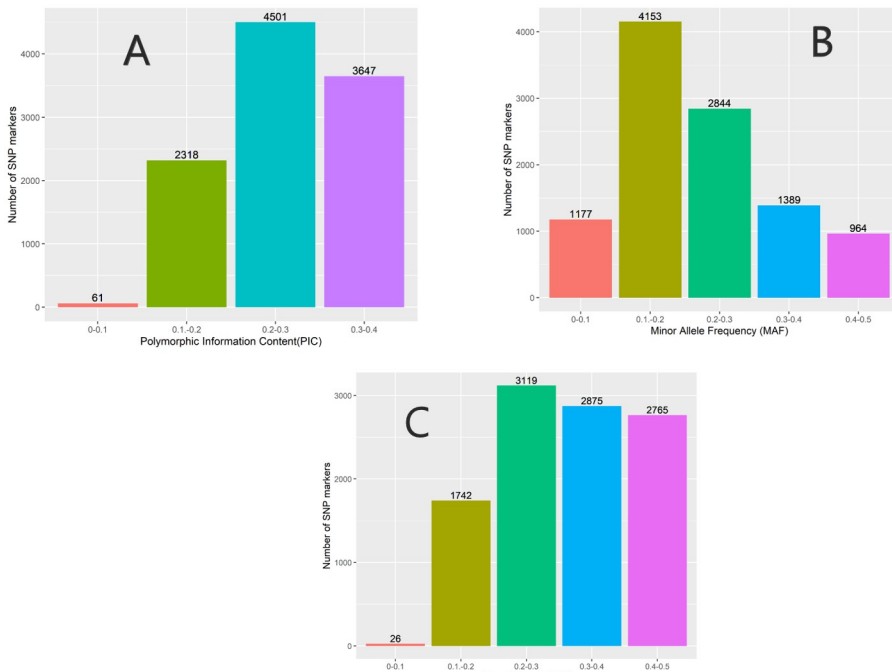

**Fig 1. Single nucleotide polymorphism markers characteristics: A = Polymorphic Information Content (PIC), B = Minor allele frequency (MAF) and C = Gene diversity (GD).**

analysis (PCoA), which showed that clusters 1, 3, and 4 showed genetic lineage, thus making between-population variance less than within-population variance ("Fig 6").

## Genetic grouping of tepary bean

Neighbor-joining hierarchical clustering assorted the population into five clusters with different numbers of genotypes as follows: Cluster 1 (24 genotypes), Cluster 2 (8 genotypes), Cluster 3 (13 genotypes), Cluster 4 (11 genotypes), and Cluster 5 (22 genotypes).

The landraces G50 and G45 were clustered with released varieties and breeding lines in Clusters 1 and 5, respectively ("Fig 7"). Most genotypes sourced from Mexico were grouped in Clusters 2, 3, 4 and 5. Similarly, genotypes from USDA were primarily grouped in Cluster 1 ("Fig 7").

## Discussion

A well-characterized genetic resource is useful for breeding and genetic analysis. Genetic analysis using molecular markers complements phenotypic selection to develop heterotic groups and genotype selection for breeding. Genetic gains for yield, grain quality, and biotic and abiotic stress tolerance depend on the magnitude of the response to selection. However, limited genomic resources are developed for tepary bean for genetic differentiation and selection. Previously, germplasm characterization in this crop has been based on the use of molecular markers developed for common bean [19,26]. Therefore, the present study aimed at deducing the genetic groups in tepary bean germplasm collection using high-density DArT-based single nucleotide polymorphism (SNPs) markers and selecting contrasting genotypes for breeding.

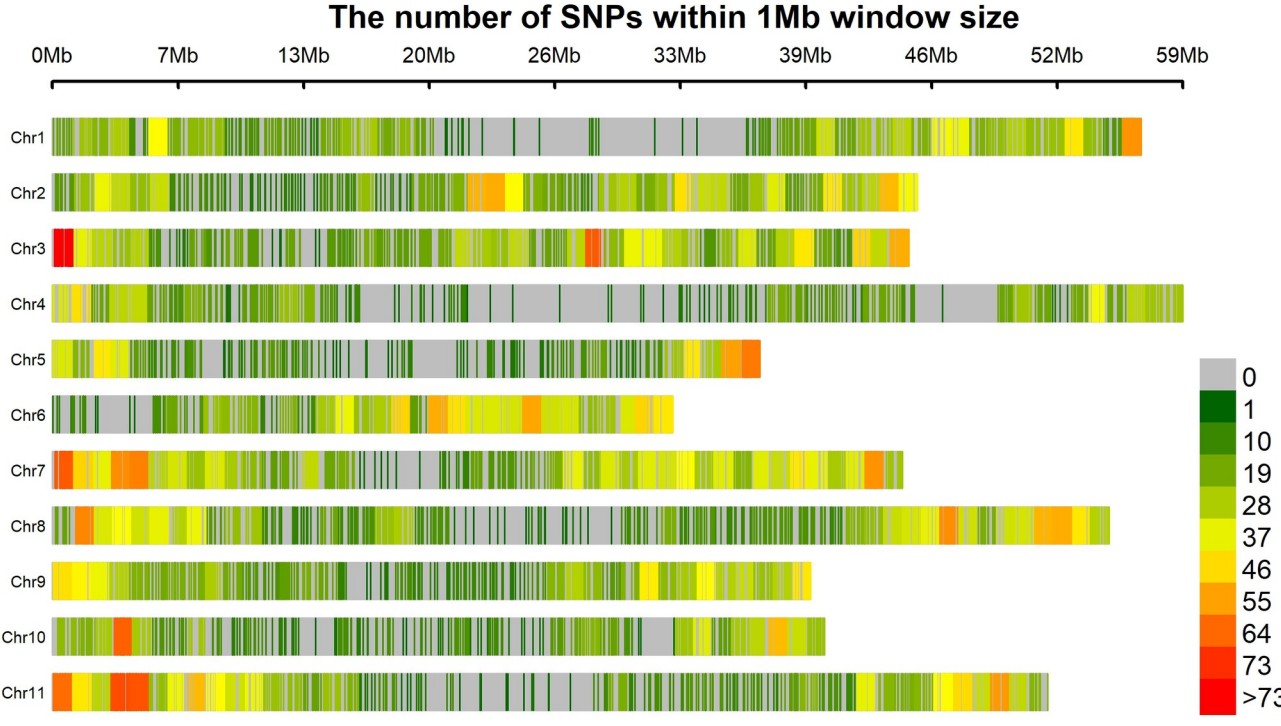

**Fig 2. Single nucleotide polymorphism density plot mapped in 78 tepary bean genotypes.**

In the present study, the least number of the derived SNPs markers recorded PIC values ranging between 0 to 0.2 and were considered less informative ("Fig 1"). The low PIC value could be attributed to the bi-allelic nature of the SNP markers, which limits them to values of less than or equal to 0.5 [50,51]. The majority of the developed SNPs markers recorded PIC values ranging from 0.2 to 0.3, and 0.3 to 0.5 ("Fig 1"). These SNPs markers are considered informative. The SNPs markers revealed moderate genetic diversity in the studied tepary bean germplasm ("Table 2"). The AMOVA ("Table 4") indicated fewer variations between populations than within the population and supplemented the moderate genetic diversity.

The moderate genetic divergence typical of cultivated tepary bean in our study corroborates with the study findings of Blair et al. [18], Gujaria-Verma et al. [26], and Mhlaba et al. [19]. This is attributed to localized domestication and distribution of the crop in the USA and

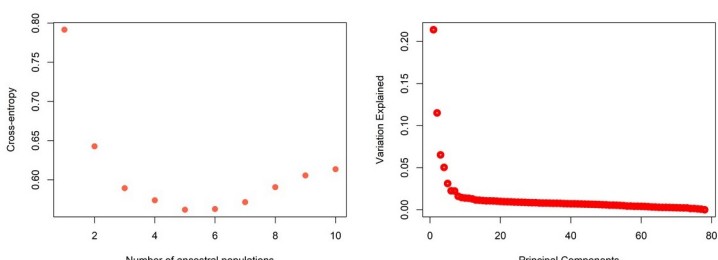

**Fig 3. Tepary bean population structure based on cross-entropy (left) and principal component analysis (right) employing 10527 high-density SNP markers.**

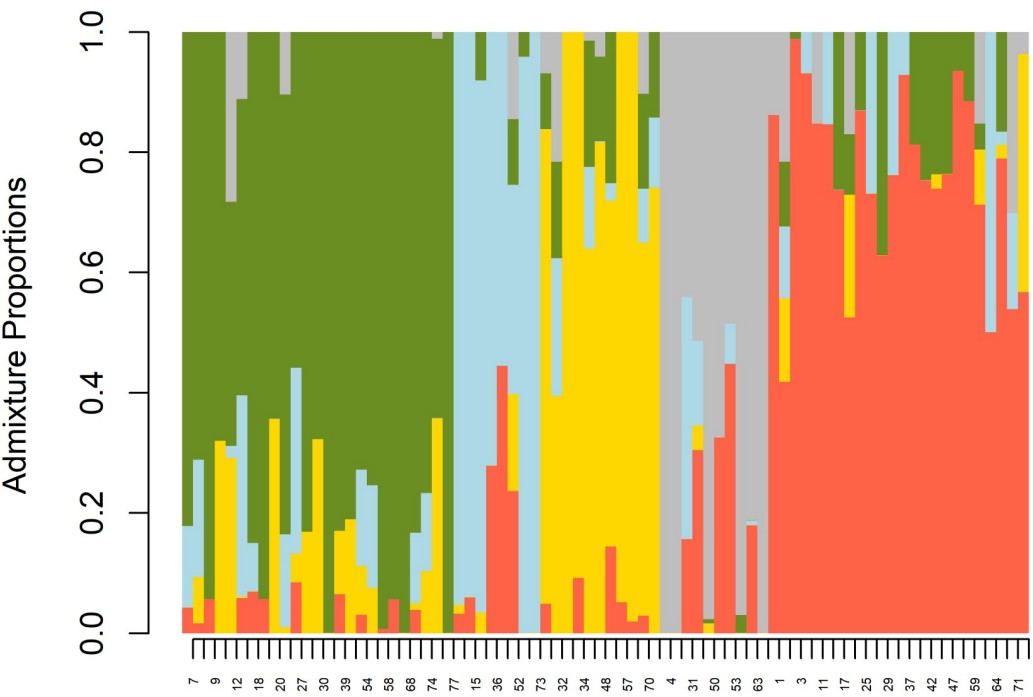

**Fig 4. Population admixtures in 78 tepary bean genotypes using 10527 high-density SNP markers.**

Mexico, which culminated in a narrow genetic base. The low natural hybridization and outcrossing rates in tepary bean could also be attributed to its moderate genetic diversity [18,19]. However, the existing genetic diversity within the studied population ("Table 4") implies that sufficient genetic variation is present for crop improvement. Moghaddam et al. [1] also reported high within-population variation in tepary bean, concurring with the present findings. Farmers and breeders may have selected tepary bean genotypes for certain agronomic traits, resulting in considerable genetic differentiation among populations [52].

The observed and expected heterozygosity were marginally different ("Table 2"), suggesting the preponderance of homozygous alleles in dominant and recessive forms at many loci in the assessed tepary bean collection. The high additive variance supports this compared to the dominance variance observed in the present study. The higher tendency favouring homozygosity is expected as the tepary bean is a self-pollinated crop with a limited outcrossing rate [13,18]. The occurrence of both dominant and recessive alleles has implications for selection and breeding. Specifically, the upregulation of dominant alleles may enhance the selection of well-adapted genotypes, while the same may hinder the selection of recessive alleles [32]. The moderate fixation index value or coefficient of inbreeding ("Table 3") suggests high gene exchange emanating from artificial and natural hybridization, which culminates in moderate genetic differentiation. Fixation indices are classified as low ($< 0.05$), moderate ($0.05–0.15$), and high ($> 0.15$) and show the level of genetic divergence or similarity in the assessed population [53]. Therefore, strategic crossing for the development of new varieties should target genetically divergent genotypes selected in different clusters ("Fig 7"). These can enhance

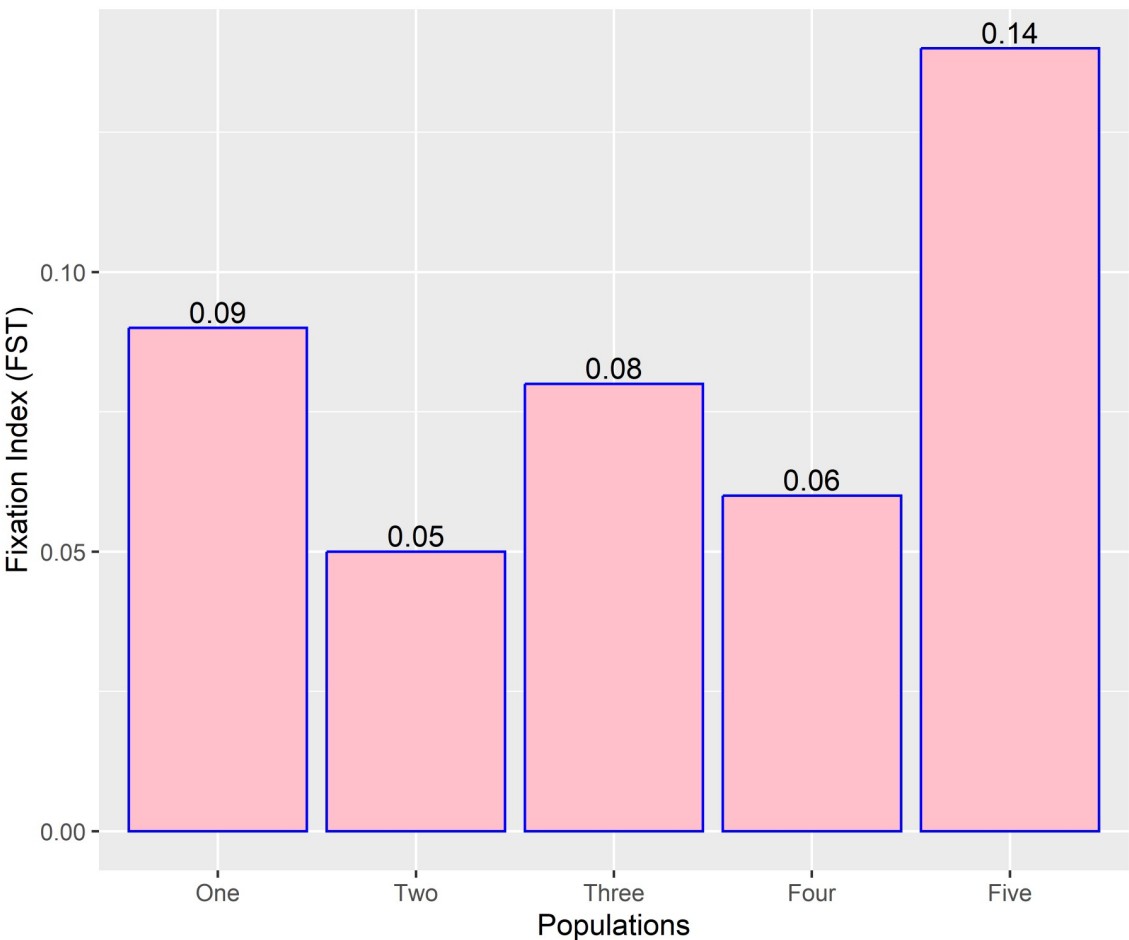

**Fig 5. Fixation index (FST) of five tepary bean subpopulations determined from population structure analysis based on SNP markers.**

genetic gains for economic traits, including grain yield, nutrient compositions, heat and drought tolerance, and disease resistance.

The clustering or grouping of tepary bean genotypes into five sub-populations following their geographical origins in the USA and Mexico reiterated the single domestication theory, which apparently caused a genetic bottleneck and limited genetic variation in tepary bean [21]. The grouping of landraces like Uchokwane (G50) and Zimbabwe landrace (G45) with released and breeding varieties in Clusters 1 and 2 signifies the presence of admixtures that could have

**Table 3. Population pair-wise fixation index of 78 tepary bean genotypes genotyped using SNPs markers.**

| Subpopulations | Subpopulation 1 | Subpopulation 2 | Subpopulation 3 | Subpopulation 4 | Subpopulation 5 |
|---|---|---|---|---|---|
| Subpopulation 1 | 1 | 0.21 | 0.08 | 0.11 | 0.23 |
| Subpopulation 2 | | 1 | 0.18 | 0.14 | 0.20 |
| Subpopulation 3 | | | 1 | 0.09 | 0.21 |
| Subpopulation 4 | | | | 1 | 0.06 |
| Subpopulation 5 | | | | | 1 |

**Table 4. Analysis of molecular variance in tepary bean germplasm collection based on SNP markers.**

| Source of variation | DF | SS | MS | Variance Estimated | % variance |
|---|---|---|---|---|---|
| Between population | 4 | 138098.70 | 34524.68 | 2097.93 | 38.12% |
| Within population | 73 | 248584.00 | 3405.26 | 3405.26 | 61.87% |
| Total | 77 | 386682.80 | 5021.85 | 5503.19 | 100% |

DF, degrees of freedom; SS, sum of squares; MS, mean square.

arisen due to historical exchanges of seeds through the informal seed system or the exchange of germplasm between breeding programs. The number of subpopulations in our study was slightly lower than the six subpopulations reported by Bornowski et al. [35]. This could be attributed to differences in the diversity panels and genotyping platform used.

The following genotypes are recommended for new variety design: Tars-Tep 112, Tars-Tep 10, Tars-Tep 23 selected from Cluster 1, G40022, Tars-Tep-93 and Tars-Tep -100 from Cluster 3, Zimbabwe landrace, G40017, G40143 and G40150 from Cluster 5. Genotypes, including Tars-Tep 23 and G40150, have high yield potential and tolerate drought and heat [54,55]. G40022 is tolerant to salt [56]. G40150 is resistant to common bacterial blight (CBB) while Tars-Tep 23 has broad resistance to common bacterial blight and rust [54,57].

## Conclusion

The present study appraised SNP markers and determined their usefulness in assessing the extent of genetic diversity and population structure in tepary bean germplasm collections. The SNP markers revealed moderate genetic diversity in the germplasm collection. The study

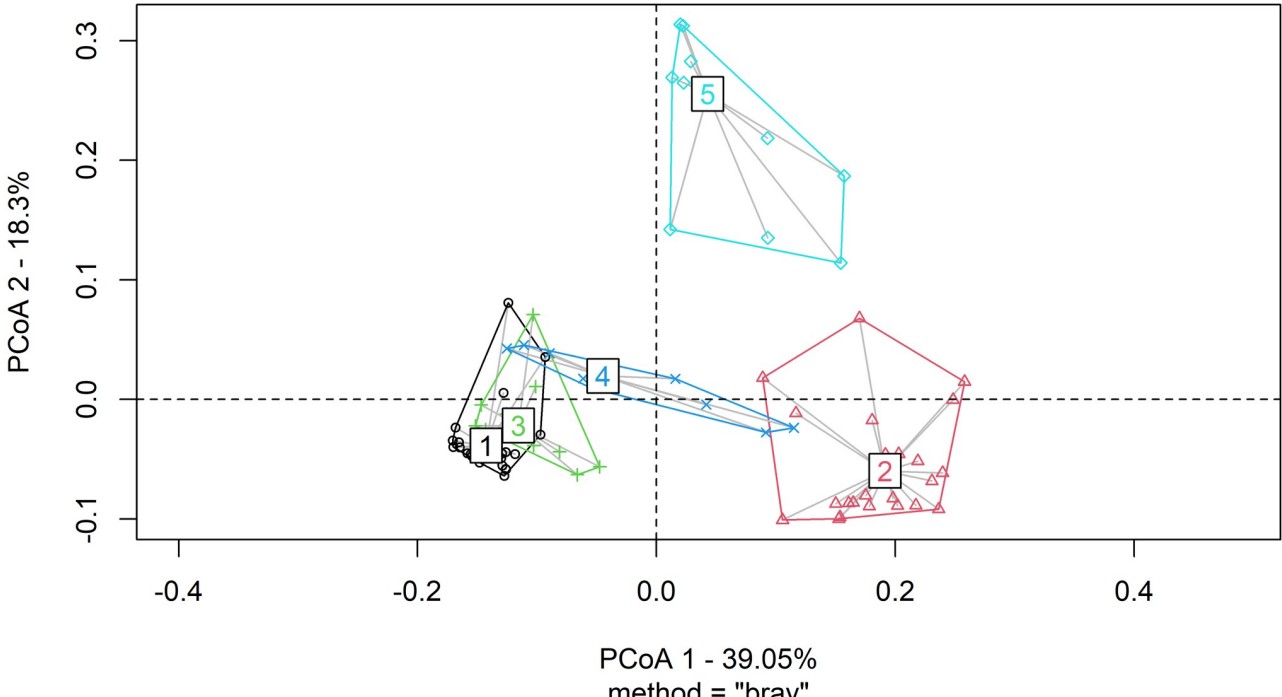

**Fig 6. Principal coordinate analysis (PCoA) of tepary bean genotypes based on population structure using SNPs markers.**

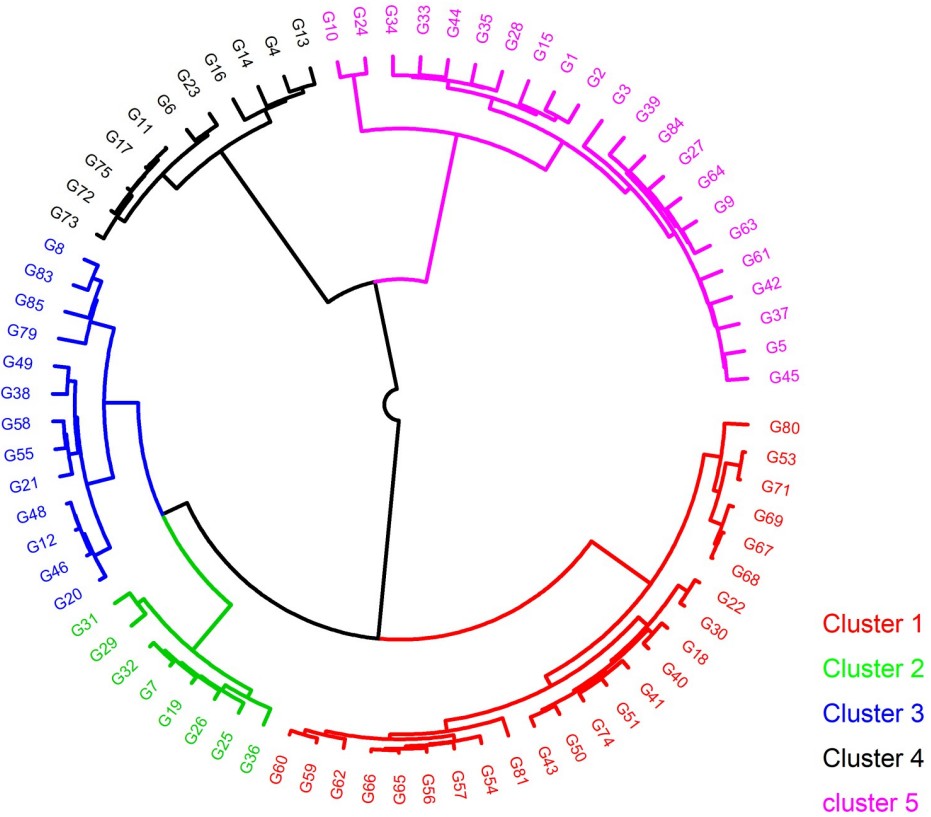

**Fig 7. Dendrogram showing the five genetic clusters (Cluster 1 to 5) among 78 tepary bean genotypes based on SNPs markers.** See codes of genotypes in "Table 1".

identified five distinctive sub-populations that were genetically differentiated. This will guide genotype selection and subsequent crosses to develop breeding populations and enhance genetic gains for economic traits. The following distantly related genotypes were selected, namely: Tars-Tep 112, Tars-Tep 10, Tars-Tep 23, Tars-Tep-86, Tars-Tep-83, and Tars-Tep 85 from Cluster 1, G40022, Tars-Tep-93, and Tars-Tep-100 from Cluster 3, Zimbabwe landrace, G40017, G40143, and G40150 from Cluster 5. The selected and contrasting accessions are valuable genetic resources to initiate crosses to enhance genetic variation and integrate traits in tepary bean and genetically related legume crops.

## Supporting information

**S1 Table. Hapman file for DArT- based SNP markers.**
(CSV)

**S2 Table. Divgenos file.**
(CSV)

## Acknowledgments

The authors would like to express their deepest gratitude to the staff of SEQART Africa for the Dartseq genotyping service and to the CIAT-Malawi staff for processing the seed samples.

## Author Contributions

**Conceptualization:** Saul Eric Mwale, Hussein Shimelis, Jacob Mashilo.

**Data curation:** Saul Eric Mwale.

**Formal analysis:** Saul Eric Mwale, Wilfred Abincha.

**Funding acquisition:** Saul Eric Mwale.

**Investigation:** Saul Eric Mwale.

**Methodology:** Saul Eric Mwale, Wilfred Abincha.

**Resources:** Saul Eric Mwale.

**Software:** Saul Eric Mwale, Wilfred Abincha.

**Supervision:** Hussein Shimelis.

**Validation:** Saul Eric Mwale, Hussein Shimelis, Wilfred Abincha, Wilson Nkhata, Abel Sefasi, Jacob Mashilo.

**Visualization:** Saul Eric Mwale, Hussein Shimelis, Wilfred Abincha, Jacob Mashilo.

**Writing – original draft:** Saul Eric Mwale.

**Writing – review & editing:** Saul Eric Mwale, Hussein Shimelis, Wilfred Abincha, Wilson Nkhata, Abel Sefasi, Jacob Mashilo.

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
