## [Decision Letter · Decision Letter 0]

6 Oct 2023

PONE-D-23-28131Genetic differentiation of tepary bean (Phaseolus acutifolius) germplasm collection using genotyping-by-sequencing with high-density Single Nucleotide Polymorphism markersPLOS ONE

Dear Dr. Mwale,

Thank you for submitting your manuscript to PLOS ONE. After careful consideration, we feel that it has merit but does not fully meet PLOS ONE’s publication criteria as it currently stands. Therefore, we invite you to submit a revised version of the manuscript that addresses the points raised during the review process.Nov 20 2023 11:59PM. If you will need more time than this to complete your revisions, please reply to this message or contact the journal office at plosone@plos.org. Please include the following items when submitting your revised manuscript:A rebuttal letter that responds to each point raised by the academic editor and reviewer(s). You should upload this letter as a separate file labeled 'Response to Reviewers'.A marked-up copy of your manuscript that highlights changes made to the original version. You should upload this as a separate file labeled 'Revised Manuscript with Track Changes'.An unmarked version of your revised paper without tracked changes. You should upload this as a separate file labeled 'Manuscript'.If applicable, we recommend that you deposit your laboratory protocols in protocols.io to enhance the reproducibility of your results. Protocols.io assigns your protocol its own identifier (DOI) so that it can be cited independently in the future. For instructions see: https://journals.plos.org/plosone/s/submission-guidelines#loc-laboratory-protocols. Additionally, PLOS ONE offers an option for publishing peer-reviewed Lab Protocol articles, which describe protocols hosted on protocols.io. Read more information on sharing protocols at https://plos.org/protocols?utm_medium=editorial-email&utm_source=authorletters&utm_campaign=protocols.

We look forward to receiving your revised manuscript.

Kind regards,

Aditya Pratap

Academic Editor

PLOS ONE

Journal Requirements:

The Kirkhouse Trust SCIO is sincerely thanked for the financial support to genotype the Tepary bean germplasm collection. Further, the African Center of Excellence in Neglected and Underutilized Biodiversity is also sincerely thanked for the financial support towards the completion of this manuscript. The authors would like to express their deepest gratitude to the staff of SEQART Africa for the Dartseq genotyping service and to the CIAT-Malawi staff for processing the seed samples.

Yes. 

Kirkhouse Trust SCIO provided the funding for the genotyping of tepary bean germplasm.

3, In your Data Availability statement, you have not specified where the minimal data set underlying the results described in your manuscript can be found. PLOS defines a study's minimal data set as the underlying data used to reach the conclusions drawn in the manuscript and any additional data required to replicate the reported study findings in their entirety. All PLOS journals require that the minimal data set be made fully available. For more information about our data policy, please see http://journals.plos.org/plosone/s/data-availability.

Additional Editor Comments:

I agree with the suggestions of reviewers who have recommended minor revisions. Kindly go through all suggestions and submit a revised manuscript.

Reviewers' comments:

Reviewer's Responses to Questions

**Comments to the Author**

1. Is the manuscript technically sound, and do the data support the conclusions?

Reviewer #1: Partly

Reviewer #2: Yes

2. Has the statistical analysis been performed appropriately and rigorously? 

Reviewer #1: Yes

Reviewer #2: Yes

3. Have the authors made all data underlying the findings in their manuscript fully available?

Reviewer #1: No

Reviewer #2: Yes

4. Is the manuscript presented in an intelligible fashion and written in standard English?

Reviewer #1: Yes

Reviewer #2: Yes

5. Review Comments to the Author

Reviewer #1: Dear authors,

The manuscript titled “Genetic differentiation of tepary bean (Phaseolus acutifolius) germplasm collection using genotyping-by-sequencing with high-density Single Nucleotide Polymorphism markers” provides an interesting perspective to deduce the genetic groups in tepary bean germplasm collection using GBS. The authors reported a total of 5 genetic groups. These interesting results added to different similar initiatives, will help accelerate breeding programs by offering the status of population structure in the dataset used.

In general, the authors have a logical sequence of analysis, in addition to having a high number of SNP molecular markers despite the limited number of genotypes. However, it is strongly recommended to be more descriptive in the bioinformatic and analytic methodology, and be more informative in the figures.. Also, add more references for better context and discussion. The above in order to have more clarity on how the analyzes were carried out, and thus be more effective in the reproducibility of the results in subsequent works. For all the above reasons, I recommend this manuscript for publication in Plos One with minor revision. Thus, the authors should address the following recommendations to improve some aspects before the publication.

Best regards,

I attach a PDF with comments

Reviewer #2: Work on functional annotation and validation of targeted impending traits of Tepary bean could have made the work better intricate since draft genome sequence of the bean is already available. That would shed better light on utility of the high resolution markers for selection and improvement of such traits. Reason for incorporating varieties, advance lines and landraces together rather than landraces alone for the study could be elucidated.

6. PLOS authors have the option to publish the peer review history of their article (what does this mean?). If published, this will include your full peer review and any attached files.

Reviewer #1: **Yes: **Felipe López-Hernández

Reviewer #2: No

---

## [Author Response · Author response to Decision Letter 0]

11 Nov 2023

Response to PLoS-ONE Comments 

Abstract

Reviewer comments: I suggest mentioning the method for selecting 5 sub-populations and adding punctuation signs for more clarity in the cluster list.

Response: Thank you very much for your comments. The method of selecting the 5 sub-populations has been specified in the manuscript and punctuation signs have been added to aid in clarity. The five sub-populations were identified using sparse non-negative matrix factorization (snmf), which computes an entropy criterion (Alexander & Lange, 2011). The entropy criterion helped in choosing the number of ancestral populations that explained the genotypic data. This was further validated using the elbow reference, which involved scree plot of the percentage of variation explained by each component of the PCA. The number of ancestral populations is closely linked to the number of PCA (Byun et al., 2017). 

Reference:

Alexander, D. H., & Lange, K. (2011). Enhancements to the ADMIXTURE algorithm for individual ancestry estimation. BMC Bioinformatics, 12. https://doi.org/10.1186/1471-2105-12-246

Byun, J., Han, Y., Gorlov, I. P., Busam, J. A., Seldin, M. F., & Amos, C. I. (2017). Ancestry inference using principal component analysis and spatial analysis: A distance-based analysis to account for population substructure. BMC Genomics, 18(1), 1–12. https://doi.org/10.1186/s12864-017-4166-8

Reviewer comments: I suggest adding the most recent works in the topic. For instance, Bornowski et al. in 2023 published a study on genotyping many accessions of tepary beans. On the other hand, I recommend including more references to analytic tools that can be used in this work related to population structure, etc. Additionally, to provide context on how this study relates to previously published research, I suggest including other works on population stratification using GBS in common beans such as Cortés & Blair 2018 or Dartseq as Valdisser et al 2017; or studies involving interspecific panels (Common Bean × Tepary Bean) like those by Cruz et al. in 2023.

Response: The suggested articles have been added, and our study findings have been contextualized in line with published literature. We have included DArTseq, a more specific GBS platform used in the current study, for clarity.

Reviewer comments: In Figure 3, please list each subplot and change the Eigenvalues of each PC plot to the variance explained of each. PC. 2. In figure 4 organize from largest to smallest according to each percentage of subpopulation and add the accession code at the bottom. In figure 6 add the percentage of variance explained by each component.

Response: Figure 3: We have plotted a PC scree plot of PCs against the variance, explained as suggested. However, we could not change the PC into subpopulations because, in this method, the number of PCs are the subpopulations. By looking at the scree plot, the PCs can reliably inform us of the optimum number of K.

Response: Figure 4: As suggested, we have added accession codes at the bottom of the plot. With your advice, we have refined the admixture plot, and the groups are better visible.

Response: Figure 6: We have added percentage variation on each coordinate.

Reviewer comment: R//Yes, but please add the reference to the R-package used for plotting. For example, did the authors use the CMplot function for Figure 2?

Response: We have cited all referenced R packages used, including CMplot.

Reviewer comment: Are the experiments or interventions appropriate for addressing the research question?

R// The experiments and the analytic methodology are appropriate for addressing the research question. However, I suggest providing greater clarity regarding the function used for the ancestry analysis by LEA. Was the algorithm employed for this approach SMNF (e.g., SMNF is an optimization of Structure)? The variant calling process also needs a more detailed explanation. It's unclear how this process was conducted. While they used DArTsoft, a software dedicated to DArTseq, it's worth noting that the use of this tool for GBS is not common. Typically, the main bioinformatic approach for GBS involves tools such as GATK, TASSEL, Stacks, etc.

Response: Thank you very much for your comment. We have provided more information on the function snmf as used in the LEA package to estimate the co-ancestry coefficient of the population. The current GBS was done using the DArTseq platform, which is reliably analyzed using DArTsoft 14. DArTsoft is an automated genotypic data analysis program that was developed to analyze sequence data generated from DArTseq (Gelaw et al., 2023; Adu et al., 2021; Alam et al., 2018).

Reviewer comment: Is there enough data to draw a conclusion? R// Yes, but I advert that the using of this panel for future association analysis would need major number of genotypes accessions.

Response: Thank you for the comment. We will consider including several accessions in future association analysis.

Reviewer comment: Do the authors follow best practices for reporting? R// No. I suggest including the company name and country when reporting the use of kits and software.

Response: We have included the company for the TANBEAD Plant Extraction Kit (Diagnocine, Hackensack, NJ, USA) and a QIAquick PCR purification kit (QIAGEN GmbH, Hilden, Germany).

Reviewer comments: Could another researcher reproduce the study with the same methods? In other words, have the authors provided enough information to validate the study? R// Partially, because the authors have not provided with a link to SNP data or the scripts used in R or other software

Response: Supporting information on genotypic data has been provided.

Reviewer comments: Do the results support the conclusions? Yes, but I recommend specifying some analysis. In that sense, from data science, the visualization of the first two principal components does not suggest an optimal number of components (PCs) but rather it is a clustering validity analysis that suggests the optimal number of clusters.

Response: Thank you for the comment. The cross-entropy criteria was used to identify the optimum number of K populations and the identified subpopulations were validated by principal component analysis.

---

## [Editor Report · Decision Letter 1]

29 Nov 2023

Genetic differentiation of a southern African tepary bean (Phaseolus acutifolius A Gray) germplasm collection using high-density DArTseq Single Nucleotide Polymorphism markers

PONE-D-23-28131R1

Dear Dr. Mwale,

We’re pleased to inform you that your manuscript has been judged scientifically suitable for publication and will be formally accepted for publication once it meets all outstanding technical requirements.

Kind regards,

Aditya Pratap

Academic Editor

PLOS ONE
---

## [Editor Report · Acceptance letter]

4 Dec 2023

PONE-D-23-28131R1 

Genetic differentiation of a southern African tepary bean (*Phaseolus acutifolius* A Gray) germplasm collection using high-density DArTseq SNP markers 

Dear Dr. Mwale:

I'm pleased to inform you that your manuscript has been deemed suitable for publication in PLOS ONE. Congratulations! Your manuscript is now with our production department. 

Kind regards, 

on behalf of

Dr. Aditya Pratap 

Academic Editor

PLOS ONE